# Effects of Exercise on Patients with Obstructive Sleep Apnea: A Systematic Review and Meta-Analysis

**DOI:** 10.3390/ijerph191710845

**Published:** 2022-08-31

**Authors:** Jiale Peng, Yuling Yuan, Yuanhui Zhao, Hong Ren

**Affiliations:** School of Sport Science, Beijing Sport University, Beijing 100084, China

**Keywords:** obstructive sleep apnea, exercise, randomized controlled trials, meta-analysis

## Abstract

With exercise being more frequently utilized in treatment for obstructive sleep apnea (OSA), a systematic review of the intervention efficacy of exercise on OSA is necessary. PubMed, EBSCO, Web of Science, VIP, and CNKI databases were searched to collect randomized controlled trials (RCTs) of exercise applied to OSA from January 2000 to January 2022. The literature screening, data extraction, and risk of bias assessment of included studies were conducted independently by two reviewers. Meta-analysis was then performed using Rev Man 5.4 software. A total of 9 RCTs were included, including 444 patients. Compared with the control group, exercise made an improvement in apnea–hypopnea index (AHI) [MD = −6.65, 95% CI (−7.77, −5.53), *p* < 0.00001], minimum oxygen saturation (SaO_2min_%) [MD = 1.67, 95% CI (0.82, 2.52), *p* = 0.0001], peak oxygen uptake (VO_2peak_) [SMD = 0.54, 95% CI (0.31, 0.78), *p* < 0.00001], Pittsburgh sleep quality index (PSQI) [MD = −2.08, 95% CI (−3.95, −0.21), *p* = 0.03], and Epworth Sleepiness Scale (ESS) values [MD = −1.64, 95% CI, (−3.07, −0.22), *p* = 0.02]. However, there were no significant changes in body mass index (BMI). As for the results of subgroup analysis, aerobic exercise combined with resistance exercise [MD = −7.36, 95% CI (−8.64, −6.08), *p* < 0.00001] had a better effect on AHI reduction than aerobic exercise alone [MD = −4.36, 95% CI (−6.67, −2.06), *p* = 0.0002]. This systematic review demonstrates that exercise reduces the severity of OSA with no changes in BMI, and the effect of aerobic exercise combined with resistance training is better than aerobic exercise alone in AHI reduction. Exercise also improves cardiopulmonary fitness, sleep quality, and excessive daytime sleepiness.

## 1. Introduction

Obstructive sleep apnea (OSA) is a common sleep disorder characterized by repeated episodes of apnea or hypopnea during sleeping. Common symptoms of OSA include irregular snoring, morning headache, excessive daytime sleepiness, and memory loss, which seriously affects people’s daily work and life [1]. The prevalence of OSA increases with age [2]. It is also related to gender and weight; it is almost several times higher in men than in women [3] and in obese people compared to people of normal weight [2,4]. OSA emerged as a public health issue that has affected nearly 1 billion people worldwide [5].

Besides the symptoms mentioned above, numerous adverse cardiovascular effects could be caused by OSA. There is a causal relationship between OSA and the incidence and morbidity of hypertension, coronary heart disease, arrhythmia, heart failure, and stroke [6]. When controlled for age, obesity, and smoking, OSA is an independent risk factor for hypertension. According to cross-sectional observations, the prevalence of arterial hypertension in subjects with OSA ranges from 35% to 80% [7,8], and a dose–response association has been found between OSA and essential hypertension [9]. There is also a dose–response relationship between the severity of OSA and the likelihood of developing atrial fibrillation, which indicates that OSA is also linked to an elevated risk of atrial fibrillation [10]. Moreover, OSA is also connected to several cardiovascular disease (CVD) risk factors. A prior study of the relationship between the occurrence of OSA and heart rate variability showed that a higher apnea–hyponea index (AHI) constitutes an independent predictor of reduced heart rate variability regarding both sympathetic and parasympathetic components and the sympathetic–parasympathetic balance [11]. Cardiovascular fitness is known as a predisposing factor for CVD [12]. An earlier investigation addressed that a decrease in percent predicted peak oxygen uptake of 3.20 was related to each one-unit increase in a log-transformed AHI [13]. The relationship between the coexistence of cardiovascular risk factors and the incidence and severity of OSA was assessed by a previous study, and a directly proportional relationship between the number of cardiovascular risk factors and AHI was shown [14]. Since OSA is a modifiable CVD risk factor, more emphasis should be put on the treatment of OSA.

To date, there is no radical cure for OSA, and comprehensive treatment is still the primary treatment method. Continuous positive airway pressure (CPAP), the most common clinical treatment method, is effective in improving AHI and daytime sleepiness [15]. However, poor compliance, nevertheless, limited its effects. CPAP withdrawal usually leads to a rapid recurrence of OSA and a return of subjective sleepiness [16].

Physical activity and exercise have long been recognized as key determinants of health [17,18] and are especially helpful in preventing chronic diseases [19]. OSA frequently coexists with other chronic diseases, and they share many risk factors; exercise may potentially be an efficacious and cost-effective method for OSA. According to a previous meta-analysis, exercise has been linked to considerable increases in cardiorespiratory fitness, daytime drowsiness, and sleep efficiency, as well as a reduction in OSA severity [20,21,22]. Additionally, a large population-based prospective cohort study demonstrated that walking and vigorous-intensity physical activity were associated with a decreased risk of OSA independent of other risk factors [17]. Exercise was shown to be the second most effective treatment for OSA behind CPAP in terms of reducing AHI in those with diagnosed OSA [23]. In recent years, a few new RCTs have demonstrated the effects of exercise on patients with OSA. A systematic review of these new research studies is necessary. Building on the previous meta-analysis, this study aimed to assess the impact of exercise on people with OSA and update it to include several additional RCTs which substantially increased the sample size (and therefore confidence in the effect size of the intervention). In particular, we were interested in determining the mean changes in AHI with different modalities of exercise that had never been discussed before to provide guidance for future exercise interventions for OSA.

## 2. Materials and Methods

This systematic review adhered to the Preferred Reporting Items for Systematic Reviews and Meta-analysis (PRISMA) statement and guidelines [24].

The systematic review analyzed the effect of exercise on AHI, body mass index (BMI), minimum oxygen saturation, daytime sleepiness, and sleep quality in adults with OSA. PubMed, EBSCO, Web of Science, VIP, and CNKI databases were electronically searched to collect RCTs of exercise applied to OSA from January 2000 to January 2022. Key terms and free words were used for screening potential studies and traced back existing references for a supplement. The retrieval strategies are shown in the Appendix A.

Studies eligible for inclusion had to fulfill the following criteria: (1) randomized controlled trials, (2) adults (age ≥ 18 years) who conformed to the diagnostic criteria of OSA (AHI ≥ 5 events per hour) with no treatment (CPAP/oral appliance/nasal surgery), (3) control group received regular treatment (health guidance/stretching), while intervention group had additional exercise, and (4) studies included results on AHI, BMI, and some other related parameters, such as SaO_2min_%, peak oxygen uptake, daytime sleepiness, and sleep quality. Studies were excluded from meta-analysis if: (1) the study was duplicate published, (2) the data could not be extracted, (3) the study was not published in English or Chinese, and (4) it had patients with cardiovascular, respiratory, or psychiatric disease.

Two reviewers screened literature independently; the steps included importing all references into Endnote, checking for duplicates, and screening the titles and abstracts to remove irrelevant studies, then reading the full text of potential studies according to eligibility criteria. If there were any different opinions, a discussion would be organized, and if the opinions of two reviewers were not consistent, a third person would be consulted.

The quality of studies was evaluated using the Cochrane risk of bias tool which is a valid tool in the field of meta-analysis. Seven dimensions, random sequence generation, allocation concealment, blinding of participants and personnel, blinding of outcome assessment, incomplete outcome data, selective reporting, and other biases were included in the tool, and each dimension was rated as “high risk”, “low risk”, or “unclear” [25]. If there was any disagreement during the preliminary evaluation, the two reviewers would have a discussion first. If no agreement could be reached, a third person would be consulted.

Data extraction was carried out using the pre-established data extraction table by two reviewers independently. The following information was extracted: title, author, published time, characteristics of patients, exercise program, key elements of risk of bias, and outcomes. AHI was the primary outcome of this systematic review, whereas SaO_2min_%, peak oxygen uptake, daytime sleepiness, and sleep quality were the secondary outcomes.

Meta-analysis was performed using Rev Man 5.4 software (The Nordic Cochrane Center, The Cochrane Collaboration, Copenhagen, Denmark ). Mean difference (MD) or standard mean difference (SMD) were used as effect analysis statistics for measurement data, and each effect size provided 95% confidence interval (CI). The heterogeneity among the included studies was analyzed by *χ*^2^ test (α = 0.1), and the heterogeneity was quantitatively determined by *I*^2^. If there was no statistical heterogeneity among the results, the fixed-effect model was used for meta-analysis; otherwise, the source of heterogeneity was further analyzed, and the random-effect model was used for meta-analysis after excluding the influence of obvious heterogeneity. If there was significant heterogeneity in each study, methods such as subgroup analysis or sensitivity analysis were used. The level of the meta-analysis was set as *α* = 0.05.

## 3. Results

### 3.1. Results of the Search

A total of 516 references were electronically retrieved, and 1 study was manually searched. After deleting duplicate studies, 278 studies were obtained. A total of 249 publications were excluded after screening titles and abstracts, and the full text of 29 studies that were potentially suitable for inclusion was reviewed. Finally, 9 studies [26,27,28,29,30,31,32,33,34] met the eligibility criteria, including 444 patients. The main reasons for exclusion were described in the flowchart, which is shown in Figure 1.

### 3.2. Characteristics of the Studies

The characteristics of the included studies are summarized in Table 1, Table 2 and Table 3. Four studies [27,28,29,34] had protocols with aerobic exercise, and five studies [26,30,31,32,33] had protocols with aerobic exercise combined with resistance exercise. All the exercise interventions followed the principle of exercise prescription, regardless of the exercise modality. Various types of aerobic exercise protocols were applied in the articles, including exercising on a treadmill or cycle ergometer, brisk walking outside, performing traditional Chinese exercises such as taijiquan, etc. In most studies, low-to-moderate-intensity aerobic exercise was preferred by researchers, and the intensity of the exercise was monitored by heart rates or rating of perceived exertion (RPE). The total duration of exercise included in the studies was about an hour, including warm-up for 5 to 15 min, formal exercise for 30 to 40 min, and stretching for 5 to 15 min. Resistance training, targeting most of the muscle groups, was performed with light weights and multiple repetitions. Exercise frequency ranged from three to six times per week, and the total intervention period ranged from 4 to 36 weeks. The Cochrane risk of bias evaluation results are presented in Figure 2.

### 3.3. Meta-Analysis Results

Nine studies [26,27,28,29,30,31,32,33,34] showed a significant pooled estimate of mean change in AHI of −6.65 events/h (95% CI −7.77 to −5.53; *p* < 0.00001) with an *I*^2^ = 39%. All studies were divided into two groups according to the modality of exercise for subgroup analysis, and the results show that aerobic exercise alone [27,28,29,34] contributed to a mean change in AHI of −4.36 events/h (95% CI −6.67 to −2.06; *p* = 0.0002) with an *I*^2^ = 0%, while aerobic exercise combined with resistance exercise [26,30,31,32,33] contributed to a mean change in AHI of −7.36 events/h (95% CI −8.64 to −6.08; *p* < 0.00001) with an *I*^2^ = 40%. The difference between subgroups was statistically significant (*p* = 0.03) (Figure 3).

Six studies reported BMI [27,28,29,30,32,34]. There was no significant difference between exercise group and control group after intervention [MD = −0.57, 95% CI (−1.44, 0.30), *p* = 0.20] with an *I*^2^ = 47% (Figure 4A).

Five studies [26,28,29,30,33] reported SaO_2min_%. Exercise was found to significantly improve SaO_2min_% after intervention [MD = 1.67, 95% CI (0.82, 2.52), *p* = 0.0001] with an *I*^2^ = 34% (Figure 4B).

Six studies [26,29,30,31,32,34] reported peak oxygen uptake. Exercise was found to significantly improve peak oxygen uptake after intervention [SMD = 0.54, 95% CI (0.31, 0.78), *p* < 0.00001] with an *I*^2^ = 14% (Figure 4C).

Four studies [27,28,32,34] reported daytime sleepiness conditions. Exercise was found to significantly reduce Epworth scale values after intervention [MD = −1.64, 95% CI (−3.07, −0.22), *p* = 0.02] with an *I*^2^ = 0% (Figure 4D).

Three studies [28,32,33] reported quality of sleep. Exercise was found to significantly decrease PSQI after intervention [MD = −2.08, 95% CI (−3.95, −0.21), *p* = 0.03] with an *I*^2^ = 85% (Figure 4E).

## 4. Discussion

This meta-analysis examined the effect of exercise interventions on OSA-related health outcomes. One of the main findings of this meta-analysis is that the total pooled estimate of mean change in AHI was reduced by 5.95, slightly lower than the results of previous meta-analyses [21,35]. This difference may be explained by the fact that the participants included in this study were mostly mild-to-moderate OSA patients, whereas those with other chronic disease complications were also excluded. However, there were no significant changes in BMI noted. The findings indicate that the decrease in AHI may be independent of weight loss. As for the mechanism of how exercise improves OSA, evidence was shown in previous studies. Some researchers suggested that exercise could decrease AHI due to changes in the muscle tone of the upper respiratory tract [36]. When individuals inhale, the capacity of contraction of the dilator muscle tone of the upper respiratory tract is the key to maintaining airway patency. In patients with OSA, the loss of pharyngeal dilator muscle tone at sleep triggers recurrent pharyngeal collapse and temporary cessation of breathing (apnea) [37]. Previous studies suggested that long-term regular exercise might promote upper airway muscle activation to increase upper airway diameter, reduce airway resistance, and oppose pharyngeal collapse during sleeping [36,38], and this promotion reduces the incidence of hypopnea or apnea during sleeping. As a result, exercise’s ability to lower AHI and raise SaO2min% may be related to enhanced upper respiratory tract muscle tone.

Unimpeded venous return is important to maintain health, while a sedentary lifestyle will impair the venous return and consequently cause fluid retention in the legs. However, OSA patients appeared to be engaging in less physical activity and more sedentariness [17]. Due to the recumbent position of the patient as they fell asleep in bed, the fluid retention in the legs moved to the neck and pooled there, increasing laryngeal compression [36]. This process most likely has a role in the increased AHI. Exercise promoted the backflow of venous blood and reduced fluid retention in the legs. Therefore, improvement in the severity of OSA due to exercise is probably attributed to reduced fluid retention in the legs, which consequently reduces the fluid volume shifted from the leg to the rostral when sleeping and results in a dilatation of the upper airway. Research supported the idea above that a decreased AHI due to exercise is associated with a greater reduction in the overnight change in leg fluid volume [39]. In contrast with previous studies, this study analyzed the different intervention effects of two exercise modalities. An examination of subgroups revealed a significant difference between the effects of aerobic exercise alone and aerobic exercise combined with resistance training on AHI reduction. These findings are likely connected to the fact that resistance training was more effective at boosting leg muscle strength. Leg muscles play a central role in venous reflux, which helps to improve the retention of muscular fluid in the legs.

Cardiorespiratory fitness refers to the capacity of the circulatory and respiratory systems to supply oxygen to skeletal muscle mitochondria for energy production needed during physical activity [40]. It is an important predictor of cardiovascular disease, all-cause mortality, and mortality rates attributable to various cancers [12]. OSA had impaired cardiorespiratory fitness, which was manifested by the decrease in peak oxygen uptake and poor oxygen uptake efficiency [13]. It was associated with adverse cardiovascular diseases, such as hypertension, atherosclerosis, and coronary heart disease. These cardiovascular consequences of OSA were largely mediated by chronic intermittent hypoxia and sleep fragmentation, and the pathomechanism primarily involved sympathetic activity, oxidative stress, and inflammation [41]. The results of this meta-analysis suggested that regular exercise’s ability to effectively increase the peak oxygen uptake of OSA may be attributed to the improvement in the primary molecular mechanisms mentioned above. It indeed has been shown in research that exercise can improve the autonomic nervous function of OSA by balancing the tension of sympathetic and parasympathetic nerves, thus playing a certain role in cardiovascular protection [30]. However, existing research has not determined the effects of exercise on inflammation or oxidative stress yet.

The most common complaints among individuals with OSA are excessive daytime sleepiness, fatigue, tiredness, lack of energy, and poor sleep quality [42,43]. The Epworth Sleepiness Scale was used to measure the level of daytime sleepiness. Patients with more severe daytime sleepiness were characterized by higher ESS scores [44]. In this meta-analysis, a significant improvement in daytime sleepiness and quality of sleep was observed. These findings are consistent with a previous meta-analysis [21]. There are certain advantages of exercise over CPAP, even if the effects on OSA with exercise demonstrated in this study appear to be less favorable than those achieved with CPAP [45]. On the one hand, exercise requires less time commitment. The duration of the exercise program included in this study was about an hour, while CPAP therapy often lasts more than four hours. CPAP is not always well tolerated, and nonadherence is its major limitation [46]. On the other hand, CPAP therapy was unfriendly to the patients who were overweight or obese because they seemed to increase BMI further by using CPAP therapy alone [47], while exercise did not result in significant changes in BMI in this study. Moreover, despite optimal CPAP compliance, one in five patients who utilized CPAP for more than 7 h per night still had abnormal ESS values [48]. As our study did not provide enough evidence, it remains to be verified whether exercise or CPAP is more beneficial in reversing abnormal ESS values in patients with OSA.

Our research had some strengths. First, the most recent studies on OSA and exercise were included in this meta-analysis. Second, compared to the previous meta-analysis, we included more participants with 444 individuals. Third, there was low heterogeneity between the experimental and control groups for the exercise-related changes in AHI, BMI, SaO2min%, VO2peak, and ESS. Finally, we determined the mean changes in AHI with two modalities of exercise and compared their difference. There were also some limitations to this study. When this study discussed the influence of exercise on the sleep quality of OSA, the result showed high heterogeneity, which needs to be supplemented by more randomized controlled studies for subgroup discussion. Furthermore, this meta-analysis did not analyze the efficacy of different exercise intensities or duration of exercise on OSA severity. More studies need to be included to clarify these effects in the future.

## 5. Conclusions

This systematic review demonstrates that exercise can reduce the severity of OSA with no changes in BMI, and the effect of aerobic exercise combined with resistance training is better than aerobic exercise alone in AHI reduction. Exercise also can improve cardiopulmonary fitness, sleep quality, and excessive daytime sleepiness conditions. Given the limitations of existing research, more high-quality, large-scale, multi-center, long-term randomized controlled trials need to be conducted to provide clinical evidence.

## Figures and Tables

**Figure 1 ijerph-19-10845-f001:**
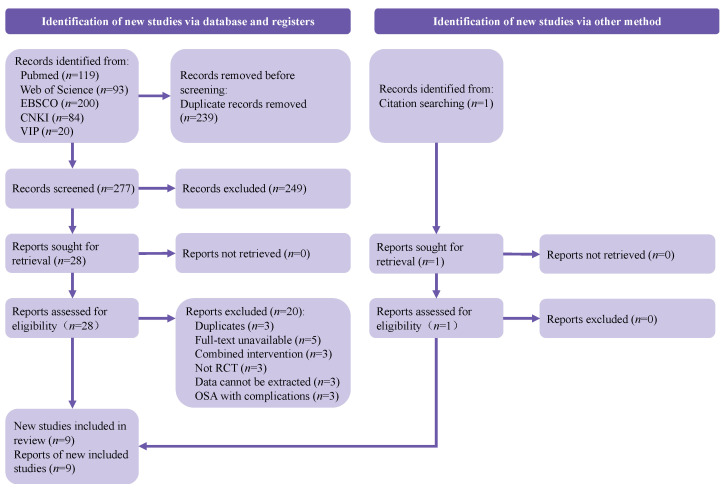
PRISMA diagram.

**Figure 2 ijerph-19-10845-f002:**
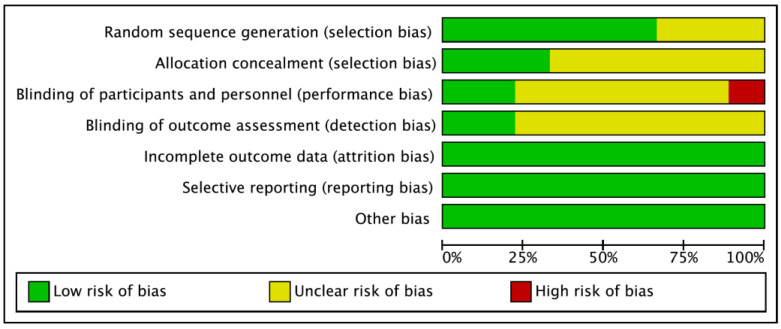
Risk of bias in studies.

**Figure 3 ijerph-19-10845-f003:**
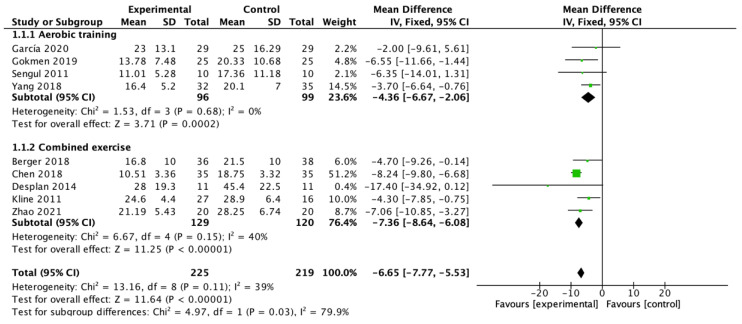
Meta-analysis of the pooled effect of exercise training on AHI and sub-analysis of the pooled effect of aerobic training and combined exercise training on AHI [26,27,28,29,30,31,32,33,34].

**Figure 4 ijerph-19-10845-f004:**
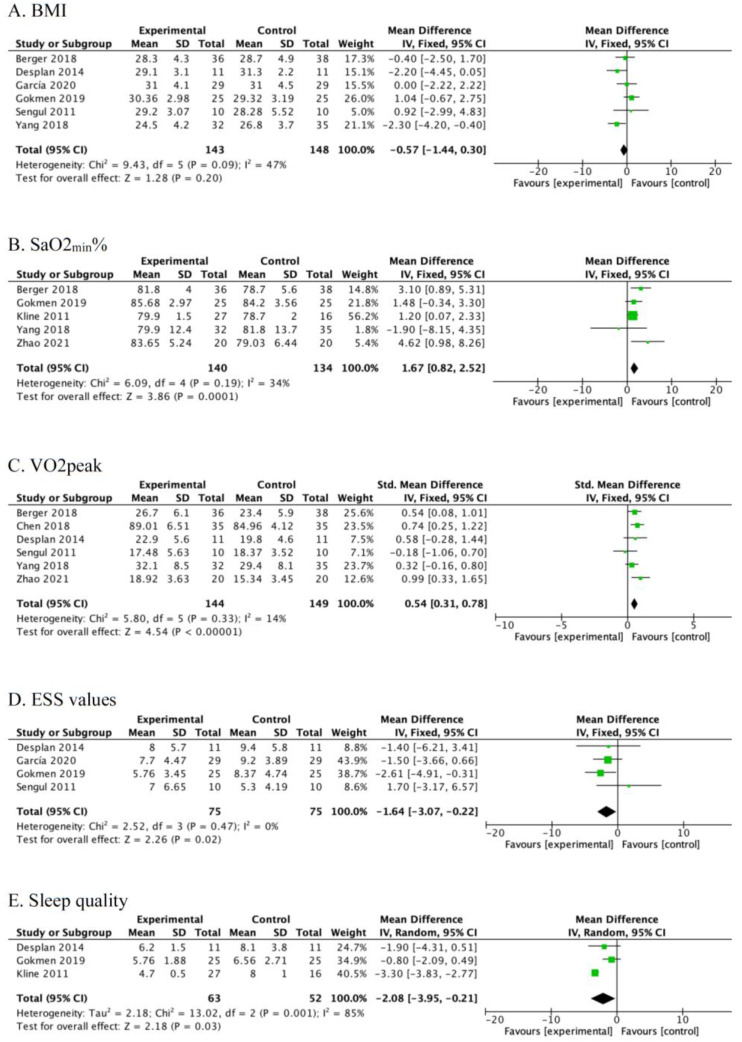
Meta-analysis of the pooled effect of exercise training on BMI (**A**), SaO2min% (**B**), VO2peak (**C**), ESS values (**D**), and sleep quality (**E**) [26,27,28,29,30,31,32,33,34].

**Table 1 ijerph-19-10845-t001:** Characteristics of studies.

Author /Year	Age		Sample Size		BMI (kg/m^2^)		AHI (Times/Hour)		Outcome
	Training	Control	Training (Male)	Control (Male)	Training	Control	Training	Control	
ZHAO /2021 [26]	49.61 ± 4.35	48.93 ± 3.62	20 (13)	20 (15)	28.87 ± 3.90	29.56 ± 4.25	27.55 ± 6.12| 21.19 ± 5.43	26.90 ± 6.48| 28.25 ± 6.74	①②⑤
García /2020 [27]	52 ± 6.6	50 ± 9.5	34 (20)	34 (23)	32.0 ± 4.1| 31.0 ± 4.1	32 ± 4.3| 31 ± 4.5	29.0 ± 20.8| 23.0 ± 13.1	27.0 ± 9.9| 25.0 ± 16.29	①③
Gokmen /2019 [28]	50.44 ± 8.38	45.68 ± 7.64	25 (13)	25 (18)	30.56 ± 2.99| 30.36 ± 2.98	29.21 ± 3.49| 29.32 ± 3.19	19.32 ± 7.09| 13.78 ± 7.48	18.66 ± 6.14| 20.33 ± 10.68	①②③④
Yang /2018 [29]	46.3 ± 6.4	48.6 ± 7.2	32 (22)	35 (24)	27.6 ± 4.7| 24.5 ± 4.2	27.1 ± 3.5| 26.8 ± 3.7	20.2 ± 7.5| 16.4 ± 5.2	19.5 ± 6.1| 20.1 ± 7.0	①②⑤
Berger /2019 [30]	60~64	60~65	36 (24)	38 (22)	28.4 ± 4.3| 28.3 ± 4.3	28.5 ± 4.5| 28.7 ± 4.9	21.9 ± 7.0| 16.8 ± 10.0	21.0 ± 6.3| 21.5 ± 10.0	①②⑤
CHEN /2018 [31]	47.4 ± 7.3	48.3 ± 7.9	35 (20)	35 (23)	25. 8 ± 3. 5	26.4 ± 3.9	25.13 ± 2.44| 10.51 ± 3.36	24.74 ± 2.49| 18.75 ± 3.32	①⑤
Desplan /2014 [32]	35~70		13	13	29.9 ± 3.4| 29.1 ± 3.1	31.3 ± 2.5| 31.3 ± 2.2	40.6 ± 19.4| 28.0 ± 19.3	39.8 ± 19.2| 28.0 ± 19.3	①③④⑤
Kline /2011 [33]	47.6 ± 1.3	45.9 ± 2.2	27 (15)	16 (9)	105.6 ± 3.0 (kg)| 104.7 ± 3.1 (kg)	99.3 ± 5.1 (kg)| 98.7 ± 5.0 (kg)	32.2 ± 5.6| 24.6 ± 4.4	24.4 ± 5.6| 28.9 ± 6.4	①②④
Sengul /2011 [34]	54.40 ± 6.57	48.0 ± 7.49	10 (10)	10 (10)	29.79 ± 2.66| 29.20 ± 3.07	28.42 ± 5.42| 28.28 ± 5.52	15.19 ± 5.43| 11.01 ± 5.28	17.92 ± 6.45| 17.36 ± 11.18	①③⑤

Before “|” is before intervention; after “|” is after intervention; ① AHI, ② SaO2min%, ③ ESS, ④ PSQI, ⑤ VO2peak.

**Table 2 ijerph-19-10845-t002:** Origin country and inclusion/exclusion criteria of studies.

Author/Year	Country	Inclusion/Exclusion Criteria
ZHAO/2021 [26]	China	Inclusion criteria: (1) moderate-to-serious OSA patients with no medication or surgery treatment; (2) ages 44 to 45; (3) physically inactive. Exclusion criteria: (1) serious cardiopulmonary or metabolic diseases; (2) uncontrolled high blood pressure (>155/99 mmHg); (3) arrhythmia; (4) bone, muscle, nervous system, and other diseases unable to complete the exercise evaluation and treatment; (5) long-term use of hypnotic or sedative medications; (6) systemic serious organic disease; (7) electrolyte disorder.
García/2020 [27]	Spain	Inclusion criteria: (1) moderate OSA patients or serious OSA patients with CPAP refusal; (2) ages 18 to 65; (3) physically inactive. Exclusion criteria: (1) impairment of the locomotor system; (2) respiratory failure caused by chronic cardiopulmonary disease; (3) severe psychiatric illness preventing the understanding of and/or compliance with instructions.
Gokmen/2019 [28]	Turkey	Inclusion criteria: (1) mild to moderate OSA patients with no treatment (CPAP, oral devices, nasal surgery, tennis ball/positional therapy, diuretic, etc.); (2) ages 30 to 65; (3) physically inactive; (4) BMI ≤ 35 kg/m^2^. Exclusion criteria: (1) taking hypnotic or sedative medications; (2) morphological defect (facial malformation, etc.), which can cause sleep disorders; (3) smoking or alcoholism; (4) having orthopedic, neurological, or musculoskeletal problems, which impede exercising; (5) pregnant women; (6) uncompensated clinical conditions such as chronic obstructive pulmonary disease, interstitial pulmonary disease, heart failure, or rheumatic and psychiatric illnesses.
Yang/2018 [29]	China	Inclusion criteria: (1) newly diagnosed mild-to-moderate OSA with CPAP refusal, none of whom received surgical or mechanical ventilation treatment prior to inclusion. Exclusion criteria: (1) diabetes mellitus, hypertension, coronary artery disease, peripheral arterial disease, thyroid disorder, severe ventricular arrhythmia, severe reduction in LVEF, valvular disease requiring surgery, severe renal dysfunction, severe orthopedic problems that would prohibit exercise, history of psychiatric or neurodegenerative disorders, acute systemic illness, or circadian desynchrony (e.g., shift workers).
Berger/2019 [30]	France	Inclusion criteria: (1) moderate OSA patients without treatment; (2) ages 40 to 80. Exclusion criteria: (1) cardiovascular or respiratory comorbidities; (2) excessive daytime sleepiness justifying immediate initiation of CPAP; (3) respiratory or heart disease contraindicating exercise discovered during stress testing; (4) Parkinson’s disease.
CHEN/2018 [31]	China	Inclusion criteria: (1) mild-to-serious OSA patients; (2) at least 18 years old. Exclusion criteria: (1) heart and lung diseases such as hypertension, coronary heart disease, chronic obstructive pulmonary disease, angina pectoris, and myocardial infarction; (2) motor disorders and serious heart, liver, and kidney organic disease.
Desplan/2014 [32]	France	Inclusion criteria: (1) a recent (<1 month) diagnosis of moderate-to-severe untreated OSA; (2) ages 35 to 70; (3) physically inactive. Exclusion criteria: (1) BMI ≥ 40 kg/m^2^; (2) regular use of hypnotic medications; (3) unstable cardiovascular disease.
Kline/2011 [33]	America	Inclusion criteria: (1) moderate-to-serious OSA patients with no treatment; (2) ages 18 to 55; (3) overweight/obese (BMI ≥ 25 kg/m^2^); (4) sedentary (<2 exercise sessions/week); (5) at stable (>3 month) medication doses (e.g., antihypertensives, antidepressants). Exclusion criteria: (1) known or suspected significant cardiovascular, pulmonary, or metabolic disease; (2) uncontrolled hypertension (>159/99 mm Hg); (3) pregnancy; (4) inability to exercise due to orthopedic or musculoskeletal problems.
Sengul/2011 [34]	Turkey	Inclusion criteria: (1) men; (2) mild-to-moderate OSA patients; (3) ages 40 to 65; (4) in good general health (stability of clinic state). Exclusion criteria: (1) angina pectoris, congestive heart failure, cardiomyopathy, emphysema, lung cancer, recent upper respiratory surgery, chronic obstructive pulmonary disease; (2) neurological, psychological, and cooperation problems that would prevent successful participation in and completion of the protocol.

**Table 3 ijerph-19-10845-t003:** Description of study intervention.

Author/Year	Intervention						
	Experimental Group						Control Group
	T1	F	I	T2	P	S	
ZHAO /2021 [26]	AE: treadmill/seat treadmill/cycle ergometer RT: the leg muscles	3	AE: 60~75% peak power RT: 3 sets of 8~10 RM	60	12	Yes	Routine health guidance (e.g., smoking cessation, healthy diet, sleep regularly, etc.).
García /2020 [27]	Walking	5	RPE: 11~15	30~50	24	No	Received general therapeutic measures, and regular physical activity monitored with a pedometer was recommended.
Gokmen /2019 [28]	Tai Chi	5	RPE: 11~13	60	12	Yes	Breathing and posture exercises (stretching).
Yang /2018 [29]	Cycle ergometer	3	Anaerobic threshold	60	12	Yes	Maintained their previous lifestyle.
Berger /2019 [30]	AE: Nordic walking/gymnastics/aqua gym RT: /	3	AE: anaerobic threshold RT: /	60	36	Yes	Received standard diet and physical activity advice.
CHEN /2018 [31]	AE: brisk walking/stretching/cycle ergometer/Tai Chi RT: dumbbells	5	AE: 60~80% VO2max RT: not mentioned	20~30	8	Yes	Routine health guidance (such as healthy diet, smoking and alcohol restriction, good sleep habit, physical activity).
Desplan /2014 [32]	AE: cycle ergometer RT: /	6	AE: ventilatory threshold HR RT: /	AE:75 RE:30	4	Yes	Outpatient standard health education program twice weekly.
Kline /2011 [33]	AE: treadmill/elliptical trainer/cycle bicycle RT: limbs, shoulder, and leg muscles	4 2	AE: 60% of HRR RT: 2 sets of 10~12 RM	45~60	12	Yes	Stretching 2 times/week for 12 weeks. At each visit, participants performed 2 sets of 12~15 stretches, each held for 15–30 s, which focused on whole body flexibility.
Sengul /2011 [34]	Treadmill/cycle ergometer	3	60~70% VO2max	45~60	12	Yes	The control group did not receive any treatment.

AE: aerobic exercise; RT: resistance training; T1: type; F: frequency (times/week); I: intensity; T2: time; P: period; S: supervised or not; RM: repetition maximum; HR: heart rate; HRR: heart rate reverse; “RT: /” indicates no resistance training.

## Data Availability

Data analyzed in this study were a re-analysis of existing data.

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
