# Peer review of "Effects of Exercise on Patients with Obstructive Sleep Apnea: A Systematic Review and Meta-Analysis"

_ijerph, 2022, doi:10.3390/ijerph191710845_

Round 1

Reviewer 1 Report

Your manuscript was very well written and interesting.

I have included some comments. Please check and correct them.

Line 64-65: PubMed search strategy is shown in the appendix, but please also include search formulas used in other databases.

Line 68: The term "oral devices" is inappropriate; please correct it to "oral appliance".

Line 105: The number of articles initially selected in the database search was listed as 516. Please describe in detail how many articles were selected in each database you used, such as PubMed, EBSCO, Web of Science, etc.

Reviewer 2 Report

The effect of exercise on obstructive sleep apnea is of considerable interest to the field of sleep medicine. Your efforts to examine the effect of aerobic exercise alone compared to aerobic and resistance exercise is novel and interesting. Unfortunately, the manuscript has some major flaws as listed below.

Major comments:

1.     The manuscript is filled with grammatical errors. There are probably more sentences with grammatical errors than ones that are grammatically correct. There are too many errors for me to list them. The manuscript should be revised by someone who has a better command of English.

2.     The findings reported in the abstract and text are inconsistent with the data. You claim that the meta-analysis shows that exercise increases minimum oxygen saturation and maximum oxygen uptake. However, Figure 4B shows the opposite i.e. improvements in minimum oxygen saturation and peak oxygen uptake favor the control group.

3.     The discussion over-interprets the findings of your study

a.     Mechanism by which exercise reduces AHI. In the first paragraph (lines 182-183), you state that “this study showed that exercise reduced AHI and increased SaO2min% was most likely relate(d) to the increased muscle tone of the upper airway”. However, you have not provided any data to suggest that this mechanism is any more or less likely than alternate mechanisms such as the effect of exercise in reducing extra-vascular fluid retention. I recommend you discuss the mechanisms by which exercise may reduce AHI but avoid making any judgements about their relative importance.

b.     Comparison of exercise and CPAP. In paragraph 3, you claim your study shows the effect of exercise was “similar to what was typically achieved in CPAP” and suggest that exercise is superior to CPAP therapy in the management of OSA based on the shorter time required for exercise compared to CPAP use, the residual sleepiness in 20% of patients adherent to CPAP and the tendency for weight gain among patients on CPAP therapy. However, the effects on OSA with exercise demonstrated in your study appear inferior to those achieved with CPAP (1) and you have not provided any evidence that exercise improves sleepiness in all patients with OSA. I think it is reasonable for you to claim that exercise has some benefits over CPAP but there is no basis for claiming that exercise is superior to CPAP for most patients with OSA.

Minor comments:

1.     The claim that OSA is twice as prevalent in obese compared to normal weight individuals is incorrect and the reference used to support this is incorrect. The prevalence of OSA in obese individuals is several times higher than normal weight individuals (2).

2.     The use of the abbreviation CRF for cardiorespiratory fitness is confusing as CRF is a common abbreviation for chronic renal failure. I suggest you either omit or change the abbreviation.

3.     In Table 4, it would be helpful to label each Forrest plot with the variable e.g. A. BMI, B. SaO2min%.

4.     In paragraph 3 of the Discussion, you refer to sympathetic activity, oxidative stress and inflammation as “primary molecular domains”. What do you mean by this?

5.     Reference 5 is incorrect. Please correct the first author.

References

1.         Patil SP, Ayappa IA, Caples SM, Kimoff RJ, Patel SR, Harrod CG. Treatment of Adult Obstructive Sleep Apnea With Positive Airway Pressure: An American Academy of Sleep Medicine Systematic Review, Meta-Analysis, and GRADE Assessment. J Clin Sleep Med. 2019;15(2):301-34.

2.         Tufik S, Santos-Silva R, Taddei JA, Bittencourt LR. Obstructive sleep apnea syndrome in the Sao Paulo Epidemiologic Sleep Study. Sleep Med. 2010;11(5):441-6.

Reviewer 3 Report

This is very interesting and well written systematic review. I found only a few minor flaws:

1. The presented systematic review has to be reported in accordance to the latest https://www.bmj.com/content/372/bmj.n71 https://prisma-statement.org/PRISMAStatement/PRISMAStatement

2. Authors have to prepare the PRISMA 2020 Flow diagram instead of Figure 1 https://prisma-statement.org/PRISMAStatement/PRISMAStatement

3. Authors have to remove all outdated references. I mean articles published before 2010.

4. The Introduction is very poor and limited. Authors wrote only a few small facts within Introduction. In introduction Authors have to highlight that OSA is related to cardiovascular disease (CVD), hypertension, and obesity based on a latest literature and then write why exercises are very important among OSA patients. Afterwards Authors have to provide why there systematic review is new and important. Authors have to provide current definitions and findings about OSA and OSA relationship with CVD and hypertension from new, relevant, and outstanding articles (please remember that CVD has been affected by COVID-19) e.g.

Benjafield AV, Ayas NT, Eastwood PR, Heinzer R, Ip MSM, Morrell MJ, Nunez CM, Patel SR, Penzel T, Pépin JL, Peppard PE, Sinha S, Tufik S, Valentine K, Malhotra A. Estimation of the global prevalence and burden of obstructive sleep apnoea: a literature-based analysis. Lancet Respir Med. 2019 Aug;7(8):687-698. doi: 10.1016/S2213-2600(19)30198-5.

Martynowicz H, Dymczyk P, Dominiak M, Kazubowska K, Skomro R, Poreba R, Gac P, Wojakowska A, Mazur G, Wieckiewicz M. Evaluation of Intensity of Sleep Bruxism in Arterial Hypertension. J Clin Med. 2018 Oct 5;7(10):327. doi: 10.3390/jcm7100327.

Urbanik D, Gać P, Martynowicz H, Poręba M, Podgórski M, Negrusz-Kawecka M, Mazur G, Sobieszczańska M, Poręba R. Obstructive Sleep Apnea as a Predictor of Abnormal Heart Rate Turbulence. J Clin Med. 2019 Dec 18;9(1):1. doi: 10.3390/jcm9010001.

Urbanik D, Gać P, Martynowicz H, Poręba M, Podgórski M, Negrusz-Kawecka M, Mazur G, Sobieszczańska M, Poręba R. Obstructive sleep apnea as a predictor of reduced heart rate variability. Sleep Med. 2019 Feb;54:8-15. doi: 10.1016/j.sleep.2018.09.014.

Gać P, Urbanik D, Macek P, Martynowicz H, Mazur G, Poręba R. Coexistence of cardiovascular risk factors and obstructive sleep apnoea in polysomnography. Respir Physiol Neurobiol. 2022 Jan;295:103782. doi: 10.1016/j.resp.2021.103782.

Mochol J, Gawrys J, Gajecki D, Szahidewicz-Krupska E, Martynowicz H, Doroszko A. Cardiovascular Disorders Triggered by Obstructive Sleep Apnea-A Focus on Endothelium and Blood Components. Int J Mol Sci. 2021 May 12;22(10):5139. doi: 10.3390/ijms22105139.

Baillieul S, Dekkers M, Brill AK, Schmidt MH, Detante O, Pépin JL, Tamisier R, Bassetti CLA. Sleep apnoea and ischaemic stroke: current knowledge and future directions. Lancet Neurol. 2022 Jan;21(1):78-88. doi: 10.1016/S1474-4422(21)00321-5.

Piątek Z, Gać P, Poręba M. The COVID-19 pandemic, heart and cardiovascular diseases: What we have learned. Dent Med Probl. 2021;58(2):219–227. doi:10.17219/dmp/133153

4. Authors have to define a precise time frame for articles searching not only write "... from inception to January, 2022".
